# Multifaceted Analysis of IL-23A- and/or EBI3-Including Cytokines Produced by Psoriatic Keratinocytes

**DOI:** 10.3390/ijms222312659

**Published:** 2021-11-23

**Authors:** Kota Tachibana, Nina Tang, Hitoshi Urakami, Ai Kajita, Mina Kobashi, Hayato Nomura, Minori Sasakura, Satoru Sugihara, Fan Jiang, Nahoko Tomonobu, Masakiyo Sakaguchi, Mamoru Ouchida, Shin Morizane

**Affiliations:** 1Department of Dermatology, Okayama University Graduate School of Medicine, Dentistry, and Pharmaceutical Science, 2-5-1 Shikata-cho, Kitaku, Okayama 700-8558, Japan; py4r2ift@okayama-u.ac.jp (K.T.); tanna@126.com (N.T.); p2ra59n8@s.okayama-u.ac.jp (H.U.); gmd421029@s.okayama-u.ac.jp (A.K.); mina584@okayama-u.ac.jp (M.K.); pnx69sky@okayama-u.ac.jp (H.N.); me422114@s.okayama-u.ac.jp (M.S.); pavl0623@okayama-u.ac.jp (S.S.); 2Department of Dermatology, Southern Medical University Zhujiang Hospital, 253 Gongye Middle Ave, Guangzhou 510280, China; 3Department of Cell Biology, Okayama University Graduate School of Medicine, Dentistry, and Pharmaceutical Science, 2-5-1 Shikata-cho, Kitaku, Okayama 700-8558, Japan; pdbq7wx4@s.okayama-u.ac.jp (F.J.); n-tomonobu@okayama-u.ac.jp (N.T.); masa-s@md.okayama-u.ac.jp (M.S.); 4Department of Molecular Oncology, Okayama University Graduate School of Medicine, Dentistry, and Pharmaceutical Science, 2-5-1 Shikata-cho, Kitaku, Okayama 700-8558, Japan; ouchidam@md.okayama-u.ac.jp

**Keywords:** psoriasis vulgaris, interleukin (IL) 23, IL-39, p19, Epstein–Barr virus-induced (EBI) 3, tildrakizumab

## Abstract

Interleukin (IL) 23 (p19/p40) plays a critical role in the pathogenesis of psoriasis and is upregulated in psoriasis skin lesions. In clinical practice, anti-IL-23Ap19 antibodies are highly effective against psoriasis. IL-39 (p19/ Epstein-Barr virus-induced (EBI) 3), a newly discovered cytokine in 2015, shares the p19 subunit with IL-23. Anti-IL-23Ap19 antibodies may bind to IL-39; also, the cytokine may contribute to the pathogenesis of psoriasis. To investigate IL23Ap19- and/or EBI3-including cytokines in psoriatic keratinocytes, we analyzed IL-23Ap19 and EBI3 expressions in psoriasis skin lesions, using immunohistochemistry and normal human epidermal keratinocytes (NHEKs) stimulated with inflammatory cytokines, using quantitative real-time polymerase chain reaction (RT-PCR), enzyme-linked immunosorbent assay (ELISA), and liquid chromatography-electrospray tandem mass spectrometry (LC-Ms/Ms). Immunohistochemical analysis showed that IL-23Ap19 and EBI3 expressions were upregulated in the psoriasis skin lesions. In vitro, these expressions were synergistically induced by the triple combination of tumor necrosis factor (TNF)-α, IL-17A, and interferon (IFN)-γ, and suppressed by dexamethasone, vitamin D3, and acitretin. In ELISA and LC-Ms/Ms analyses, keratinocyte-derived IL-23Ap19 and EBI3, but not heterodimeric forms, were detected with humanized anti-IL-23Ap19 monoclonal antibodies, tildrakizumab, and anti-EBI3 antibodies, respectively. Psoriatic keratinocytes may express IL-23Ap19 and EBI3 proteins in a monomer or homopolymer, such as homodimer or homotrimer.

## 1. Introduction

Psoriasis is a chronic inflammatory skin disease characterized by epidermal thickening and neutrophil and T-cell infiltration [1]. It is clinically characterized by well-dermacated, scaly erythematous plaques that typically develop on the scalp, elbows, knees, and buttocks [2]. Patients with psoriasis have inflammation not only on the skin, but also on the whole body, which increases the risk of various complications, such as psoriatic arthritis, cardiovascular disease, diabetes mellitus, obesity, and so on [3,4,5,6,7,8]. 

Interleukin (IL) 23 plays a critical role in the pathogenesis of psoriasis and is increased in the lesional skin of psoriasis [9,10,11,12,13]. IL-23 is a member of the IL-12 cytokine family forming heterodimers comprised of α-subunits p19 and β-subunits p40 [14]. In addition to IL-23, the IL-12 cytokine family includes IL-12 (p35/p40), IL-27 (p28/EBI3), IL-35 (p35/EBI3), and IL-39 (p19/EBI3). IL-23 drives the differentiation, proliferation, and maintenance of T helper 17 (Th17) cells [14,15,16]. Anti-IL-23Ap19 antibodies (risankizumab, tildrakizumab, guselkumab) are already known to be clinically effective against moderate-to-severe psoriasis [17,18,19]. On the other hand, p40, which constitutes IL-12 with p35 and IL-23 with p19, is also a therapeutic target for psoriasis. However, anti-IL-12/23p40 antibodies (ustekinumab) are less effective than anti-IL-23Ap19 antibodies for psoriasis [20]. Some groups reported that collateral targeting of IL-12 by anti-IL-12/23p40 antibodies is counterproductive in the therapy of psoriasis [21].

A cytokine composed of IL-23Ap19 and Epstein–Barr virus-induced (EBI) 3 heterodimer was found in transfected HEK293 cells by Ramnath et al. in 2015 [22]. The following year, the novel cytokine was named IL-39 (p19/EBI3) and was shown to be produced by B cell lymphocytes and activated neutrophils [23,24]. IL-39 was shown to mediate inflammation in lupus-like mice and act as a signal in both the IL-23R/gp130 receptor and the signal transducer and activator of transcription (STAT) 1/STAT3 pathways [23]. Anti-IL-39 polyclonal antibodies were shown to improve autoimmune symptoms in lupus-like mice [25]. In theory, the anti-IL-23Ap19 antibodies can bind to both IL-23 and IL-39, which suggests that both cytokines contribute to the pathogenesis of psoriasis, based on the clinical success of anti-IL23Ap19 antibodies against moderate-to-severe psoriasis. In addition, the existence of IL-39 in psoriatic skin might explain why anti-IL23Ap19 antibodies are more effective than anti-IL-12/23p40 antibodies. On the other hand, some reports have suggested no evidence of the existence or function of IL-39 in human and human immune cells [26,27]. Recent publications assessing the combinatorial potential of cytokine subunits in the IL-12 family failed to detect IL-39 in transfected HEK293 cells [28]. Expressions of *IL23Ap19* and *EBI3* mRNAs were detected by quantitative polymerase chain reaction (qPCR) in human epidermal keratinocytes stimulated with polycytidylic acid (Poly (I:C)) and in oral epithelial cells stimulated with IL-36γ [22,29]. However, little information is available on the production of IL-23Ap19- and/or EBI3-including cytokines, such as IL-23 (p19/p40), IL-27 (p28/EBI3), IL-35 (p35/EBI3), and IL-39 (p19/EBI3), in human epidermal keratinocytes.

Further detailed analysis of the expressions of these cytokines in the cells and their involvement in the pathogenesis of psoriasis will contribute to the elucidation of the pathophysiology of psoriasis and may unlock new therapeutic options. Therefore, in this study, we investigated IL-23Ap19- and/or EBI3-including cytokines in psoriatic keratinocytes, using transcriptional and proteomic analysis.

## 2. Results

### 2.1. The Expression of IL-23Ap19 and EBI3 Is Upregulated in Psoriatic Keratinocytes

To investigate whether human epidermal keratinocytes express IL-23Ap19 and EBI3 in psoriasis, we first immunostained normal skin and lesional psoriatic skin sections with polyclonal rabbit anti-IL-23Ap19 antibody and monoclonal mouse anti-EBI3 antibody. As shown in Figure 1, immunohistochemical staining showed IL-23Ap19 and EBI3 expression in the epidermal keratinocytes. The expressions of IL-23Ap19 and EBI3 were intra- and perinuclear in normal skin. (Figure 1A,C). However, in psoriatic skin, IL-23Ap19 and EBI3 were expressed diffusely throughout the entire epidermis in addition to strong intra- and perinuclear staining (Figure 1B,D).

### 2.2. TNF-α, IL-17A, and IFN-γ Synergistically Induce IL-23Ap19 and EBI3 Expression in NHEKs

Epidermal keratinocytes respond to TNF-α, IL-17A, and IFN-γ, which are involved in the pathogenesis of psoriasis, in part by producing inflammatory cytokines, chemokines, and antimicrobial peptides [30,31]. Chiricozzi et al. showed that keratinocytes synergistically upregulated these expressions by the combination of TNF-α and IL-17 in the cells [30]. The detected genes from keratinocytes stimulated with TNF-α and IL-17 included some of the highest expressed genes in psoriatic skin. This indicated an impressive correlation between IL-17/TNF-α-induced genes and the psoriasis gene signature. We previously reported the more synergistic activities of the combined stimulation of TNF-α, IL-17A, and IFN-γ in epidermal keratinocytes [32]. For example, the triple stimulation synergistically induces IL-17C, IL-36γ, and the antimicrobial peptide human b-defensin-2 in epidermal keratinocytes [32]. Therefore, we predicted that normal human epidermal keratinocytes (NHEKs) stimulated with the triple cytokines would mimic the condition of psoriatic keratinocytes. Epidermal keratinocytes were already reported to express *IL23Ap19* and *EBI3* mRNAs after stimulation with poly (I:C) [22].

First, we stimulated NHEKs with TNF-α, IL-17A, or IFN-γ alone to investigate whether NHEKs express the genes *IL23Ap19* and *EBI3* under conditions other than poly (I:C) stimulation. TNF-α and IL-17A but not IFN-γ significantly induced expression of the *IL23Ap19* mRNA, and TNF-α and IFN-γ, but not IL-17A induced expression of the *EBI3* mRNAs in the cells (Figure 2A,B). As expected, triple stimulation of the cytokines (TNF-α, IL-17A, and IFN-γ) maximally induced the expression of both *IL23Ap19* and *EBI3* (Figure 2C,D). Other triple-stimulation combinations, such as TNF-α/IL-17A/IFN-α, TNF-α/IL-17A/IL-27, and TNF-α/IL-17A/IL-17C, did not express *IL23Ap19* and *EBI3* mRNAs as strongly as that of TNF-α, IL-17A, and IFN-γ (Figure 2C,D). *IL12Ap35* and *IL27Ap28* were undetectable in NHEKs by qPCR in our culture systems (data not shown).

### 2.3. Topical Treatment by Dexamethasone, Vitamin D3, and Acitretin Suppress p19 and EBI3 Expression in Epidermal Keratinocytes

Topical steroids, vitamin D3 ointment, and topical and oral synthetic retinoid are clinically used as standard treatments for mild-to-moderate psoriasis. We next investigated whether the expressions of *IL23Ap19* and *EBI3* are suppressed with dexamethasone, vitamin D3, and acitretin in NHEKs. Dexamethasone significantly suppressed the expression of *IL23Ap19* mRNAs, but not *EBI3* mRNAs in the cells (Figure 3A,B). Vitamin D3 and acitretin significantly suppressed the expression of both *IL23Ap19* and *EBI3* mRNAs in a dose-dependent manner (Figure 3C–F).

### 2.4. The Analysis of IL-23Ap19 and EBI3 Protein Expressions in Epidermal Keratinocytes

We further used ELISA to examine whether NHEKs produce IL-23Ap19- and/or EBI3-including cytokines, such as IL-23 (p19/p40), IL-27 (p28/EBI3), IL-35 (p35/EBI3), and IL-39 (p19/EBI3). The expressions of IL-23Ap19 and EBI3 proteins were significantly enhanced under the triple stimulation of TNF-α, IL-17A, IFN-γ in NHEKs (Figure 4A,B). However, we were not able to detect IL-23 or IL-39 by ELISA in the stimulated cells (data not shown). IL-27 and IL-35 were also undetected (data not shown). 

To further investigate what the IL-23Ap19 and EBI3 produced by NHEKs bind to, we immunoprecipitated the supernatant of keratinocytes stimulated with the triple combination (TNF-α, IL-17A, and IFN-γ), using tildrakizumab (humanized anti-human IL-23Ap19 antibody) and rabbit anti-human EBI3 antibody, and analyzed the proteins by LC-Ms/Ms. IL-23Ap19 was appropriately detected in the protein complexes immunoprecipitated with tildrakizumab, but EBI3 and p40 were not detected (Figure 5A or Appendix A). Although the analysis system also detected miscellaneous proteins, none of the proteins were prone to be paired with IL-23Ap19. We detected EBI3 in the protein complexes immunoprecipitated with anti-human EBI3 antibody, but the IL-23Ap19 protein was not included (Figure 5B, Appendix A).

Collectively, our results suggest that human epidermal keratinocytes produced IL-23Ap19 and EBI3 proteins but not the heterodimeric cytokines IL-23, IL-27, IL-35, or IL-39.

## 3. Discussion

IL-23 and IL-39 have been reported as p19-including heterodimeric cytokines. On the other hand, IL-27, IL-35, and IL-39 are known to be EBI3-including cytokines. In this study, we focused on p19- and/or EBI3-including cytokines produced by psoriatic keratinocytes. Our immunohistochemical analysis showed that the expressions of p19 and EBI3 are increased in psoriatic keratinocytes. We also confirmed that cultured epidermal keratinocytes had elevated *IL-23Ap19* and *EBI3* mRNA and protein when stimulated with the triple combination of TNF-α, IL-17A, and IFN-γ, which mimics psoriatic inflammation. Moreover, the expressions of *IL-23Ap19* and *EBI3* were suppressed by dexamethasone, vitamin D3, and acitretin, which are commonly used for psoriasis therapy. However, our ELISA experiment and LC-Ms/Ms analyses did not detect the heterodimeric cytokines IL-23 (p19/p40), IL-27 (p28/EBI3), IL-35 (p35/EBI3), or IL-39 (p19/EBI3) in epidermal keratinocytes with triple stimulation. These findings suggest that the IL-23Ap19 and EBI3 produced by epidermal keratinocytes may not form heterodimeric cytokines IL-23, IL-27, IL-35, or IL-39 and may exist as a monomer or homopolymer, such as a homodimer or homotrimer, as shown in Figure 6.

The IL-12 family is unique in having heterodimeric cytokines, and this promiscuous pairing of α/β subunits is speculated to contribute to the dual role of inflammatory and anti-inflammatory cytokines. One of the roles of the two subunits of the IL-12 cytokine family is to promote the extracellular secretion of molecules that are difficult to secrete extracellularly [28]. In other words, IL-23Ap19 and EBI3 are not easily secreted as a monomer or homopolymer, but when they form a heterodimer, they are secreted extracellularly and can function as a cytokine. However, there seems to be a difference in the ease of secretion depending on the type of IL-12 cytokine family. In a study revisiting the combinatorial potential of cytokine subunits in the IL-12 family using HEK293 T cells co-expressing a set of each possible α/β subunit combination, IL-12 and IL-23 were extensively secreted, but IL-27 and IL-35 were poorly secreted [28]. The α-subunits p19, p28, and p35 were retained inside the cell; on the other hand, the β-subunits p40 and EBI3 could be secreted independently. Meanwhile, p40, a β-subunit of IL-12 or IL-23, is able to form a homodimer (p80) [33]. The p40 homodimer (p80) binds to an IL-12 receptor, but its role remains poorly understood, due to the lack of specific monoclonal antibodies. p28, an α-subunit of IL-27, is also called IL-30 and remains functional, even in the absence of EBI3. In particular, the p28 monomer was shown to suppress anti-allogeneic immune responses and to antagonize cytokine signaling through the gp130- and IL-6-mediated production of IL-17 and IL-10 [34,35]. From these facts, IL-23Ap19 and EBI3 may form a homodimer or remain functional even as monomers, although no report prior to this has suggested that possibility.

The expression of *EBI3* and *p19* was also demonstrated by human intestinal epithelial cells [36]. However, the cells did not co-express *p40* or *p28* that are required to form IL-23 or IL-27, respectively. Taken together with our results, the expression forms of IL-23Ap19- and/or EBI3-including cytokines in epithelial cells may be different from those in immune cells.

EBI3, discovered in B lymphocytes infected with Epstein–Barr virus in 1996, regulates cell-mediated immune responses [37]. EBI3 expression has also been described in placental syncytiotrophoblasts, endothelial cells, plasma cells, etc. EBI3 downregulation contributes to type I collagen overexpression in scleroderma skin, although there have been no reports about the association between psoriasis and EBI3 [38]. Thus, we are the first to show the association and up-regulation of EBI3 in psoriatic skin, its production by NHEKs stimulated with TNF-α, IL-17A, IFN-γ, and its suppression by dexamethasone, vitamin D3, and acitretin. Further investigation is needed to clarify the detailed roles of EBI3 in psoriatic pathology and the novel properties of EBI3. EBI3 might be a therapeutic target for psoriasis in the future.

IL-23, an essential cytokine for the pathology and the treatment of psoriasis, is mainly produced by macrophages and dendritic cells, and has a crucial role for immunity to mycobacterial and fungal infection [14,39,40]. Macrophages and dendritic cells are the primary cell types responsible for the overexpression of IL-23 in psoriatic pathogenesis [39]. On the other hand, some research groups have reported the production of IL-23 by human keratinocytes, in contrast to our results. Ehst et al. reported that IL-23 protein was enhanced upon stimulation of human keratinocytes by TNF-α plus IL-17A [41]. Li et al. showed that keratinocytes could produce IL-23 at levels sufficient to cause differentiation of IL-17A-producing T cells and skin inflammation in a mouse model [13]. Although we successfully confirmed that epidermal keratinocytes had increased expressions of IL-23Ap19 [9,10,11,12,13], we were not able to detect IL-23 in the cells with either ELISA or LC-Ms/Ms. Epidermal keratinocytes might express IL-23Ap19 as a monomer or homopolymer such as homodimer or homotrimer rather than IL-23.

The role and function of IL-39 are still not entirely clear. There are still very few reports about IL-39, and some reports even deny its existence in humans. However, it seems that HEK293 cells transfected with IL-23Ap19 and EBI3 secrete IL-23Ap19/EBI3 heterodimers [22,27]. Similar to our findings, Ecoeur et al. reported the elevated expression of *IL-23Ap19* and *EBI3* mRNAs in human keratinocytes, although they failed to detect the formation of IL-23Ap19/EBI3 heterodimers in response to stimulation of neutrophils and peripheral blood mononuclear cells [27]. They concluded that the secretion of IL-23Ap19/EBI3 complexes could be forced in human cells, although the complexes may be secreted below the lower limit of detection or have no functional role. We agree with their conclusions and believe that IL-39 is unlikely to be produced by epidermal keratinocytes and unlikely to be involved in psoriatic pathogenesis.

Anti-IL-23Ap19 antibodies are characterized by their long-lasting and highly effective nature and are not inferior to anti-human IL-17A antibodies [42]. Tildrakizumab (humanized anti-human IL-23Ap19 antibody) can correctly identify the keratinocyte-derived IL-23Ap19 protein. Anti-human IL-23Ap19 antibodies may also be more effective rather than anti-IL-12/23p40 antibodies by binding to keratinocyte-derived IL-23Ap19 protein. Elucidation of the role of the IL-23Ap19 monomer or homopolymer, such as that of a homodimer or homotrimer, might contribute more profoundly to our understanding of the pathology and potential treatments of psoriasis.

In conclusion, we demonstrated that epidermal keratinocytes produce IL-23Ap19 and EBI3 proteins but did not detect the heterodimeric cytokines IL-23, IL-27, IL-35, or IL-39 in the cells. IL-23Ap19 and EBI3 produced from epidermal keratinocytes may exist as a monomer or homopolymer, such as a homodimer or homotrimer. Further investigation of the role of keratinocyte-derived IL-23Ap19 and EBI3 is required to clarify the pathogenesis of psoriasis.

## 4. Subjects and Methods

### 4.1. Immunohistochemistry

The present study was approved by the Institutional Review Board (IRB) of Okayama University Hospital (No. 1712-018). Extracutaneous biopsy specimens obtained from psoriasis patients for diagnostic use were subjected to immunostaining. Normal skin samples were obtained from healthy volunteers at Okayama University Hospital. Written informed consent was obtained from all tissue donors according to the Helsinki Declaration. Formalin-fixed, paraffin-embedded skin samples were cut into 4 µm sections and mounted on glass slides. The slides were deparaffinized and activated with citric acid buffer (0.01 mol/L, pH 6.0; LSI Medience Corporation, Tokyo, Japan) for 5 min at 90 °C in a pressure cooker. They were then incubated with peroxidase-blocking solutions (Dako, Glostrup, Copenhagen, Denmark) for 10 min at room temperature and blocking solution (Life Technologies, Carlsbad, CA, USA) at 4 °C overnight. Next, the sections were incubated with rabbit polyclonal anti-human IL-23Ap19 antibody (MBS240347; MyBio Source, San Diego, CA, USA) or mouse monoclonal anti-human IL-27/IL-35 EBI3 antibody (MAB6456; R&D Systems, Minneapolis, MN, USA) at 4 °C overnight. Finally, the slides were incubated with goat anti-mouse/rabbit IgG antibodies conjugated to peroxidase-labeled polymer (Dako, Tokyo, Japan) for 30 min. Histochemical visualization was carried out with 3,3-diaminobenzidine (DAB) (Dako).

### 4.2. Cell Culture and Stimuli

Normal human epidermal keratinocytes (NHEKs) were obtained from Cascade Biologics/Invitrogen (catalog number C-001-5C; Portland, OR, USA) and grown in serum-free EpiLife cell culture media (Cascade Biologics/Invitrogen) containing 0.06 mM Ca^2+^ and 1× EpiLife Defined Growth Supplement (EDGS; Cascade Biologics/Invitrogen, Carlsbad, CA, USA) at 37 °C under standard tissue culture conditions. Cultures were maintained for up to nine passages in this media with the addition of 100 U/mL penicillin, 100 µg/mL streptomycin, and 0.25 µg/mL amphotericin B. NHEKs were grown in 24-well flat-bottom plates (Corning Incorporated, Corning, NY, USA). Upon reaching 70–100% confluence, cells were stimulated and/or treated with one or more of the following: IL-4 (50 ng/mL; R&D Systems, Minneapolis, MN, USA), IL-17A (50 ng/mL; R & D systems), IL-17C (50 ng/mL; R&D systems), IL-22 (50 ng/mL; R&D Systems), IL-27 (50 ng/mL; R&D Systems), IL-29 (50 ng/mL; Proteintech, Rosemont, IL, USA), TNF-α (50 ng/mL; eBioscience, San Diego, CA, USA), IFN-γ (50 ng/mL; R&D Systems), dexamethasone (10-8,10-7, or 10-6 M; Sigma-Aldrich, MO, USA), 1,25(OH)2 vitamin D3 (10-8,10-7, or 10-6 M; Sigma-Aldrich), and synthetic retinoid acitretin (10-8,10-7, or 10-6 M; Sigma-Aldrich), using phosphate-buffered saline (PBS) as a vehicle for up to 24 h. After cell stimulation/treatment, culture supernatants were collected and stored at −20 °C until analysis. RNA from NHEKs was extracted using TRIzol reagent (Invitrogen). RNA was stored at −80 °C until analysis.

### 4.3. Quantitative Real-Time PCR

Complementary DNA (cDNA) was synthesized from RNA using an iScript cDNA Synthesis Kit (BioRad, Hercules, CA, USA) according to the manufacturer’s protocol. TaqMan gene Expression Assays (Applied Biosystems ABI, Foster City, CA, USA) were used to analyze the expression of human *EBI3* (assay ID: Hs01057148_m1), *IL12A* (assay ID: Hs01073447), *IL23Ap19* (assay ID: Hs00372324_m1), and *IL27A* (assay ID: Hs00377366) according to the manufacturer’s protocol (User Bulletin #2, Applied Biosystems). Glyceraldehyde 3-phosphate dehydrogenase (GAPDH) was used as an internal control to validate the RNA for each cultured keratinocyte sample. *GAPDH* mRNA was detected using the VIC-CAT CCA TGA CAA CTT TGG TA-MGB probe with the primers 5′-CTT AGC ACC CCT GGC CAA G-3′ and 5′-TGG TCA TGA GTC CTT CCA CG-3′. Each mRNA expression was calculated as the expression relative to *GAPDH* mRNA, and all data are presented as fold changes in comparison with the control (the mean value of the non-stimulated cells).

### 4.4. Enzyme-Linked Immunosorbent Assay (ELISA)

Protein levels of EBI3 (Aviva Systems Biology, San Diego, CA, USA), IL-23Ap19 (Aviva Systems Biology), IL-23 (eBioscience, San Diego, CA, USA), IL-27 (R&D systems), IL-35 (AVIVA system biology), and IL-39 (MyBioSource, San Diego, CA, USA) in culture supernatants were measured by a commercial sandwich ELISA, according to the manufacturer’s protocol. Absorbance at 450 nm was determined using a microplate reader (SH-1000Lab; Corona Electric, Hitachinaka, Ibaraki, Japan).

### 4.5. Liquid Chromatography-Electrospray Tandem Mass Spectrometry (LC-Ms/Ms)

The collected condition media from NHEKs that were stimulated with the triplet mixture of cytokines (50 ng/mL, TNF-α, IL-17A and IFN-γ) for 48 h were treated with rabbit anti-human EBI3 antibody (119-16298; Raybiotech, Peachtree Corners, GA, USA) or humanized anti-human IL-23p19 antibody (tildrakizumab, kindly provided by Sun Pharmaceutical Industries) or neither for 16 h at 4 °C. Protein G-coated magnetic beads (Sure BeadsTM Protein G Magnetic Beads; BioRad Laboratories, Hercules, CA, USA) were then added into the individual specimens and gently mixed by a rotating procedure for another 2 h at the same temperature. Magnetic immunoprecipitation (mag-IP) that allowed the beads to absorb the protein complexes (antibody-antigen and antigen bound proteins) was performed for the treated specimens according to the conventional procedure. The collected immunoprecipitates were eluted from the beads by an acidic buffer (100 mM of Glycin-HCl buffer/pH 2.0), neutralized with 1.0 M of Tris-HCl buffer/pH 9.0, dialyzed with trypsin digestion buffer (10 mM CaCl_2_, 100 mM of ammonium bicarbonate/pH 7.8), and trypsinized overnight at 37 °C. The digested proteins were directly subjected to a shotgun-type protein identification using a nano-flow liquid chromatography–mass spectrometry apparatus (Agilent 6330 Ion Trap; Agilent Technologies, Santa Clara, CA, USA) equipped with an analytical chip (Agilent HPLC-Chip; Agilent Technologies). The resulting tandem mass spectrometry spectra of the tryptic peptides were finally analyzed using Agilent software (Spectrum Mill MS Proteomics Workbench; Agilent Technologies) with the protein database (SwissPlot) for putative Homo sapiens protein identifications. The non-specific proteins detected in common with the control were removed.

### 4.6. Statistical Analysis

Results were expressed as means ± SEMs. Student’s t-test was used to determine the significance of differences between groups. One-way analysis of variance with Tukey’s test was used to determine significance among more than two groups. The analyses were performed by GraphPad Prism 4 (GraphPad Software, San Diego, CA, USA). *p*-values < 0.05 were considered to indicate significance.

## 5. Conclusions

Epidermal keratinocytes produced both IL-23Ap19 and EBI3 proteins, but we did not detect the heterodimeric cytokines IL-23, IL-27, IL-35, or IL-39. Tildrakizumab correctly identified the keratinocyte-derived IL-23Ap19 protein. IL-23Ap19 and EBI3 produced from keratinocytes may exist as a monomer or homopolymer, such as a homodimer or homotrimer. Further investigation is required to clarify the function of keratinocyte-derived IL-23Ap19 and EBI3 in psoriasis. 

## Figures and Tables

**Figure 1 ijms-22-12659-f001:**
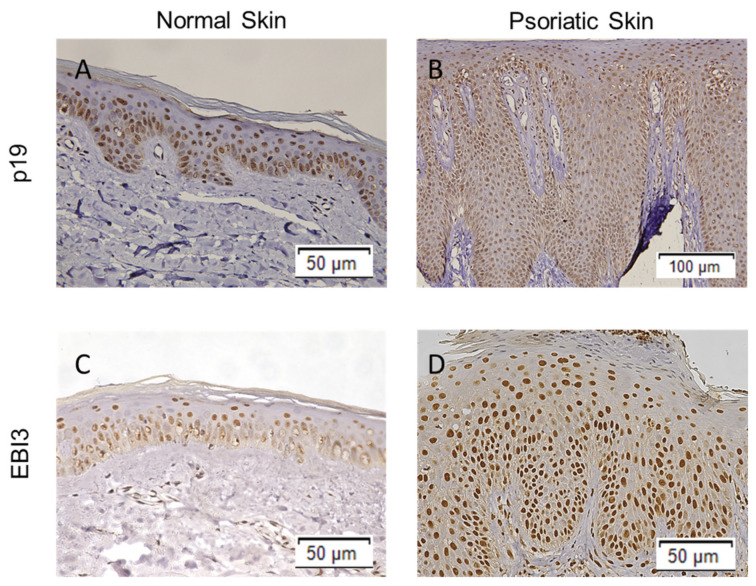
Expressions of IL-23Ap19 and EBI3 are upregulated in psoriatic keratinocytes. Expressions of IL-23Ap19 and EBI3 were examined in normal skin and lesional psoriatic skin sections by immunohistochemistry with polyclonal rabbit anti-IL-23p19 antibody (**A**,**B**) and monoclonal mouse anti-EBI3 antibody (**C**,**D**).

**Figure 2 ijms-22-12659-f002:**
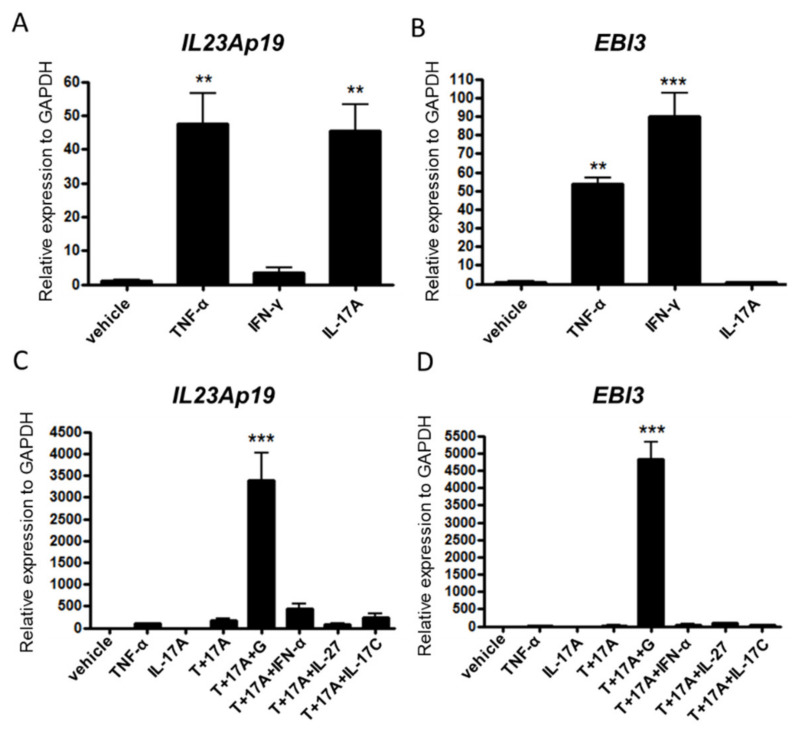
TNF-α, IL-17A, and IFN-γ synergistically induce *IL23Ap19* and *EBI3* expressions in NHEKs. (**A**–**D**) Normal human epidermal keratinocytes (NHEKs) were stimulated with (**A**,**B**) TNF-α (50 ng/mL), IL-17A (50 ng/mL), or IFN-γ (50 ng/mL), and (**C**,**D**) triple cytokine combinations of TNF-α, IL-17A, IFN-γ, IFN-α, IL-27 and/or IL-17C for 24 h. The relative mRNA expressions of *IL23Ap19* and *EBI3* to GAPDH were analyzed with quantitative real-time PCR (qPCR). ** *p* < 0.01, *** *p* < 0.001, T, TNF-α; 17A, IL-17A; G, IFN-γ.

**Figure 3 ijms-22-12659-f003:**
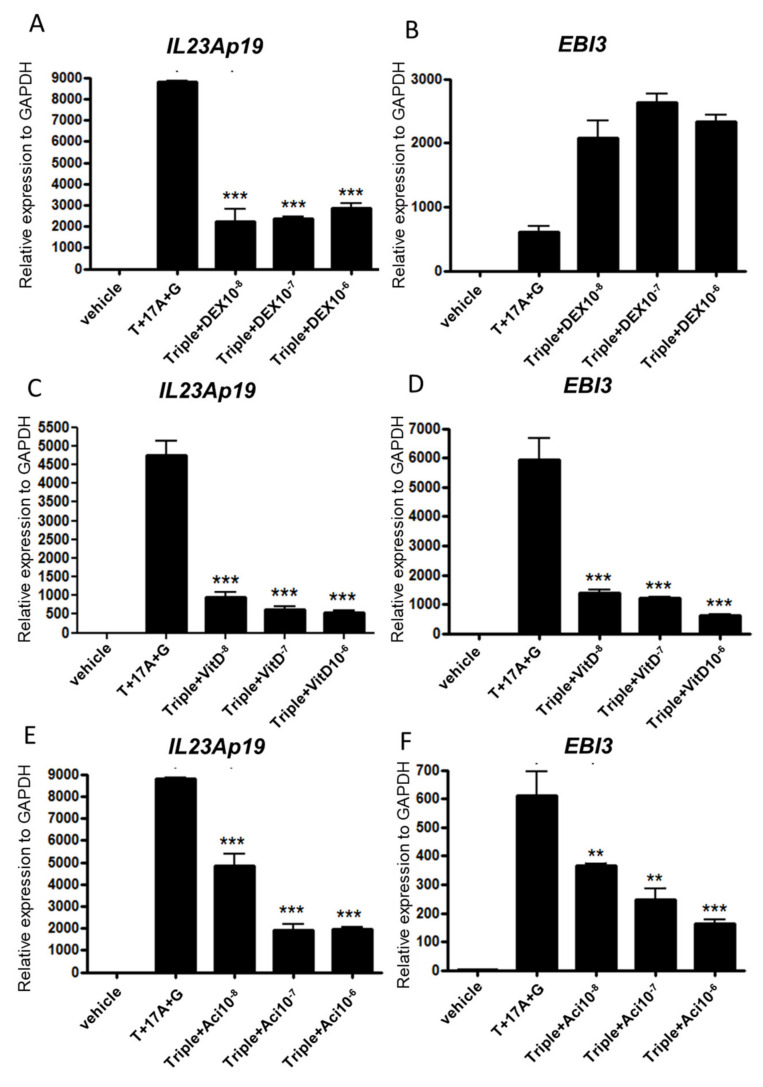
Topical treatment by dexamethasone, vitamin D3, and acitretin suppress *IL23Ap19* and *EBI3* expressions in NHEKs. Normal human epidermal keratinocytes (NHEKs) were stimulated with the triple combination of TNF-α (50 ng/mL), IL-17A (50 ng/mL), and IFN-γ (50 ng/mL), and simultaneously treated with (**A**,**B**) dexamethasone (10-8,10-7, or 10-6 M), (**C**,**D**) 1,25(OH)2 vitamin D3 (10-8,10-7, or 10-6 M), or (**E**,**F**) acitretin (10-8,10-7, or 10-6 M) for 24 h. The relative mRNA expressions of *IL23Ap19* and *EBI3* to GAPDH were analyzed with quantitative real-time PCR (qPCR). ** *p* < 0.01, *** *p* < 0.001. T, TNF- α; 17A, IL-17A; G, IFN-γ; Triple, T+17A+G; DEX, dexamethasone; VitD, 1,25(OH)2 vitamin D3; Aci, acitretin.

**Figure 4 ijms-22-12659-f004:**
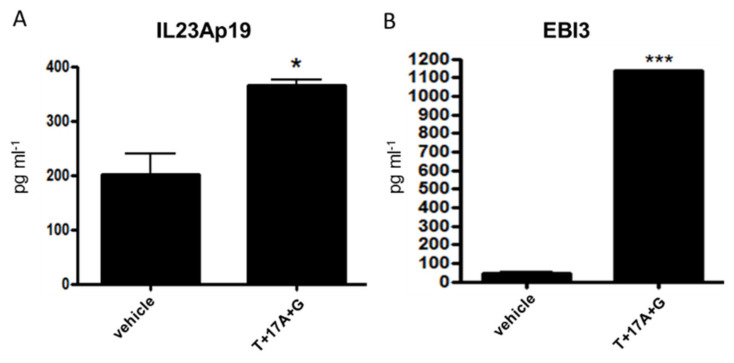
Analysis of IL-23Ap19 and EBI3 protein expressions in epidermal keratinocytes. Normal human epidermal keratinocytes (NHEKs) were stimulated with the triple combination of TNF-α (50 ng/mL), IL-17A (50 ng/mL) and IFN-γ (50 ng/mL) for 24 h. ELISA was used to measure protein levels of (**A**) IL-23Ap19 and (**B**) EBI3. * *p* < 0.05, *** *p* < 0.001. T, TNF-α; 17, IL-17A; G, IFN-γ.

**Figure 5 ijms-22-12659-f005:**
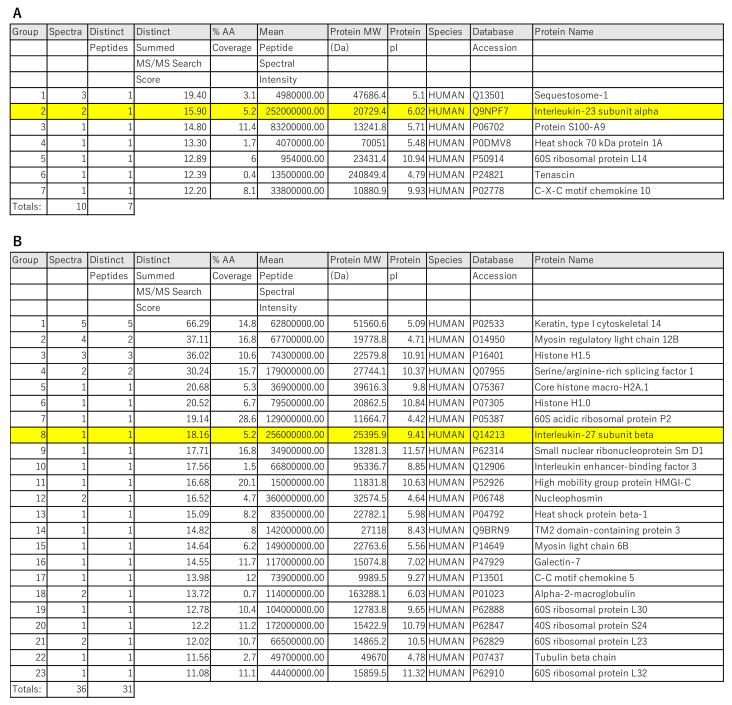
Proteins detected by immunoprecipitation with tildrakizumab or anti-human EBI3 antibody and LC-Ms/Ms. Culture supernatant of NHEKs stimulated with the triplet mixture of cytokines (50 ng/mL, TNF-α, IL-17A and IFN-γ) for 48 h were immunoprecipitated with (**A**) humanized anti-human IL-23Ap19 antibody or (**B**) rabbit anti-human EBI3 antibody and magnetic beads. The detected proteins were analyzed by liquid chromatography–electrospray tandem mass spectrometry (LC-Ms/Ms). The non-specific proteins detected in common with the control were removed.

**Figure 6 ijms-22-12659-f006:**
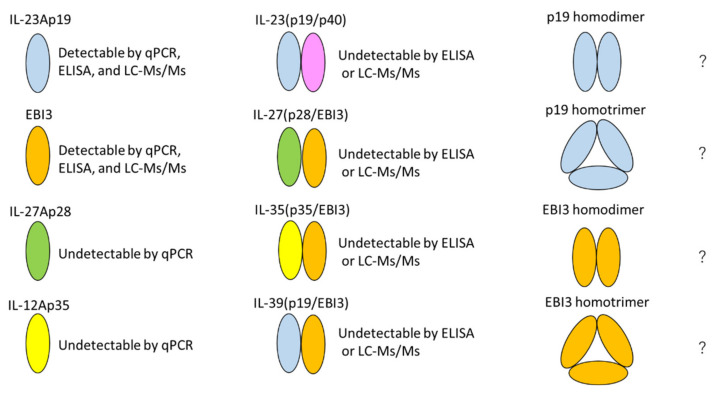
IL-23Ap19- and/or EBI3-including cytokines in epidermal keratinocytes. IL-23Ap19 and EBI3 were detected by qPCR, ELISA, and LC-Ms/Ms. IL-27Ap28 and IL-12Ap35 were undetected by qPCR. IL-23(p19/p40), IL-27(p28/EBI3), IL-35(p35/EBI3), and IL-39(p19/EBI3) were undetected by ELISA or LC-Ms/Ms. This result raises the possibility that IL-23Ap19 and EBI3 produced from epidermal keratinocytes exist as a monomer or homopolymer, such as a homodimer or homotrimer. ?, Not yet detected and possibility.

## Data Availability

The data that support the findings of this study are available from the corresponding author, upon reasonable request.

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
