# Peer review of "Multifaceted Analysis of IL-23A- and/or EBI3-Including Cytokines Produced by Psoriatic Keratinocytes"

_ijms, 2021, doi:10.3390/ijms222312659_

Round 1

Reviewer 1 Report

Congratulations for the scientific work that you have done.

I consider that the subject of your paper is verry interesting and, although the pathogenic mechanisms of psoriasis were intensively studied, they are often a challenge and still remain sometimes incompletely elucidated. 

I have a few suggestions to improve your article:

  1. I think that in section Introduction will be recommended to add a few words about clinical manifestations in psoriasis
  2. There are many others studies that refers on IL-23 and subunits p19 and p40. Your study is focused on relation between IL-23 and p19, talk also about p40 in Introduction Section
  3. In Section 2.3. you noticed the suppression of p19 and EBI3 expression induced by dexamethasone, vitamin D3, and acitretin. Please specify if you refer to topical steroid treatment or to systemic one? Include this aspect please also in the subtitle. It is necessary to note the concentration/doses, pharmaceutical forms of these products, not only the administration schedule.
  4. Please complete a Section of Conclusions
  5. Give us please more information about the medication anti-IL23A and EBI3. The theoretic and experimental part that you have been presented are more or less known and studied also by other scientific teams, I think that a discussion about tildrakizumab is appropriate now.

Author Response

Reviewer 1

Comments and Suggestions for Authors

Congratulations for the scientific work that you have done.

I consider that the subject of your paper is verry interesting and, although the pathogenic mechanisms of psoriasis were intensively studied, they are often a challenge and still remain sometimes incompletely elucidated.

I have a few suggestions to improve your article;

  1. I think that in section Introduction will be recommended to add a few words about clinical manifestations in psoriasis.

Reply: We thank Reviewer 1 for this comment. We added sentences about clinical manifestations of psoriasis in lines 44-48.

  1. There are many others studies that refers on IL-23 and subunits p19 and p40. Your study is focused on relation between IL-23 and p19, talk also about p40 in Introduction Section.

Reply: We thank Reviewer 1 for this comment. We wrote about p40 in lines 56-60.

  1. In Section 2.3. you noticed the suppression of p19 and EBI3 expression induced by dexamethasone, vitamin D3, and acitretin. Please specify if you refer to topical steroid treatment or to systemic one? Include this aspect please also in the subtitle. It is necessary to note the concentration/doses, pharmaceutical forms of these products, not only the administration schedule.

Reply: We thank Reviewer 1 for this advice. We envision topical treatment in each dexamethasone, vitamin D3, and acitretin. We included the aspect in the subtitle and line 145. Acitretin is not used for psoriasis in Japan, but in foreign countries, acitretin is used as both oral medication and topical treatment.

  1. Please complete a Section of Conclusions.

Reply: We thank Reviewer 1 for this comment. We wrote the coclusions section in lines 370-376.

  1. Give us please more information about the medication anti-IL23A and EBI3. The theoretic and experimental part that you have been presented are more or less known and studied also by other scientific teams, I think that a discussion about tildrakizumab is appropriate now.

Reply: We thank Reviewer 1 for this comment. We added the sentences in lines 56-60, and 242.

Reviewer 2 Report

This paper focuses on the expression of IL-23Ap19- and EBI3-including cytokines in psoriatic keratinocytes. To do so, they used transcriptomic and proteomic analysis on psoriatic keratinocytes (only immunohistochemistry) as well as differentially stimulated NHEKs. The results presented are interesting and innovative. However, in my opinion there is a lack of depth in the analysis to conclude on these results for publication in IJMS. It would be very interesting to use psoriatic biopsies to confirm on the proteomic results. If it is not possible, it would be relevant to elaborate further on the robustness of using the NHEK model stimulated with these cytokines and the proximity of this study model to native psoriatic skin. The paper would also benefit from a revision of the English. Here are my few other comments:

Comments:

  1. Fig 1: Results are clear for p19 but the difference between normal and psoriatic skin is not clear for EBI3.
  2. Figure legends (Fig 1, 2 and 3): The figure legends are described in a separate text under the legend. Please adjust as it was done in the legend of figure 4.
  3. Figure legends: It is not necessary to always detail the abbreviations of the cytokines in the legends of the figures since it makes the text more cumbersome. Please adjust like this example (also for the legend of figures 3 and 4):

Lines 110-115: (A-D) Normal human epidermal keratinocytes (NHEKs) were stimulated with (A, B) TNF-α (50 ng/ml), IL-17A (50 ng/ml), or IFN-γ (50 ng/ml), and (C, D) triple cytokine combinations of TNF-α, IL-17A, IFN-γ, IFN-α, IL-27 and/or IL-17C for 24 h. The relative mRNA expressions of IL23Ap19 and EBI3 to GAPDH were analyzed with quantitative real-time PCR (qPCR). **P<0.01, ***P<0.001, T, TNF-α; 17A, IL-17A; G, IFN-γ.

  1. Adjust the nomenclature of the genes in the manuscript, they should be in italics.
  2. Line 155: “Analysis of IL-23Ap19 and EBI3 protein expressions in psoriatic keratinocytes” Please change this sentence since it is in stimulated NHEK and not psoriatic keratinocytes. If it is stated like this elsewhere, please adjust.
  3. Lines 229-235: Please add references.
  4. Lines 236-240: It would be interesting to develop more on the impact of the study’s results and on the perspectives.
  5. Supplementary results: It would be interesting to add these results in the main paper (with figure 4 or after), - at least to illustrate the results that are detailed in lines 142-153, leaving the raw table in supplementary results.

Author Response

Reviewer 2

Comments and Suggestions for Authors

This paper focuses on the expression of IL-23Ap19- and EBI3-including cytokines in psoriatic keratinocytes. To do so, they used transcriptomic and proteomic analysis on psoriatic keratinocytes (only immunohistochemistry) as well as differentially stimulated NHEKs. The results presented are interesting and innovative. However, in my opinion there is a lack of depth in the analysis to conclude on these results for publication in IJMS. It would be very interesting to use psoriatic biopsies to confirm on the proteomic results. If it is not possible, it would be relevant to elaborate further on the robustness of using the NHEK model stimulated with these cytokines and the proximity of this study model to native psoriatic skin. The paper would also benefit from a revision of the English. Here are my few other comments:

Reply: We thank Reviewer 2 for these comments and suggestions. We added sentences to explain the proximity of this study model to native psoriatic skin in line 106-110.

  1. Fig 1: Results are clear for p19 but the difference between normal and psoriatic skin is not clear for EBI3.

Reply: We thank Reviewer 2 for this comment. We changed the picture to one displaying the difference between normal and psoriatic skin more clearly.

  1. Figure legends (Fig 1, 2 and 3): The figure legends are described in a separate text under the legend. Please adjust as it was done in the legend of figure 4.

Reply: We thank Reviewer 2 for this comment. We adjusted the appearance of each figure legend.

  1. Figure legends: It is not necessary to always detail the abbreviations of the cytokines in the legends of the figures since it makes the text more cumbersome. Please adjust like this example (also for the legend of figures 3 and 4):

Lines 110-115: (A-D) Normal human epidermal keratinocytes (NHEKs) were stimulated with (A, B) TNF-α (50 ng/ml), IL-17A (50 ng/ml), or IFN-γ (50 ng/ml), and (C, D) triple cytokine combinations of TNF-α, IL-17A, IFN-γ, IFN-α, IL-27 and/or IL-17C for 24 h. The relative mRNA expressions of IL23Ap19 and EBI3 to GAPDH were analyzed with quantitative real-time PCR (qPCR). **P<0.01, ***P<0.001, T, TNF-α; 17A, IL-17A; G, IFN-γ.

Reply: We thank Reviewer 2 for this advice. We deleted the abbreviations of the cytokines in figure legends of Figure2, 3, and 4.

  1. Adjust the nomenclature of the genes in the manuscript, they should be in italics.

Reply: We thank Reviewer 2 for this comment. We corrected these points.

  1. Line 155: “Analysis of IL-23Ap19 and EBI3 protein expressions in psoriatic keratinocytes” Please change this sentence since it is in stimulated NHEK and not psoriatic keratinocytes. If it is stated like this elsewhere, please adjust.

Reply: We thank Reviewer 2 for this comment. We corrected the words “psoriatic keratinocytes” to “epidermal keratinocytes”.

  1. Lines 229-235: Please add references.

Reply: We thank Reviewer 2 for this comment. We added a reference in line 269.

  1. Lines 236-240: It would be interesting to develop more on the impact of the study’s results and on the perspectives.

Reply: We thank Reviewer 2 for this comment. We added sentences to develop more on the impact of our results and the perspectives in line 277 and 279-281.

  1. Supplementary results: It would be interesting to add these results in the main paper (with figure 4 or after), - at least to illustrate the results that are detailed in lines 142-153, leaving the raw table in supplementary results.

Reply: We thank Reviewer 2 for this advice. We changed supplementary figures to Figure 5 in the main paper and added the raw data as Supplemental Figures.

Reviewer 3 Report

The paper "Multifaceted analysis of IL-23A- and/or EBI3-including cytokines produced by psoriatic keratinocytes" is a very interesting and novel study. So far there have been only a few articles published in this topic. Authors included all significant papers in the introduction and discussion sections. The study was very well designed and different methods were used to investigate p19- and/or EBI3-including cytokines in psoriatic keratinocytes. This multifaceted analysis gives very precise, not accidental results. In my opinion it would be easier to understand all levels of analysis with a graphic scheme. I am not sure if the same magnification is used in the Figure 1A, C and Figure 1B, D. Lines 127-134 should be the part of Figure 3 description. In Figure 4 there is not necessary to provide the C and D graphs.

Author Response

Reviewer 3

The paper "Multifaceted analysis of IL-23A- and/or EBI3-including cytokines produced by psoriatic keratinocytes" is a very interesting and novel study. So far there have been only a few articles published in this topic. Authors included all significant papers in the introduction and discussion sections. The study was very well designed and different methods were used to investigate p19- and/or EBI3-including cytokines in psoriatic keratinocytes. This multifaceted analysis gives very precise, not accidental results. In my opinion it would be easier to understand all levels of analysis with a graphic scheme. I am not sure if the same magnification is used in the Figure 1A, C and Figure 1B, D. Lines 127-134 should be the part of Figure 3 description. In Figure 4 there is not necessary to provide the C and D graphs.

  1. Fig 1: It would be easier to understand all levels of analysis with a graphic scheme.

Reply: We thank Reviewer 3 for this comment. We added a graphic scheme as Figure 6.

  1. I am not sure if the same magnification is used in the Figure 1A, C and Figure 1B, D.

Reply: We thank Reviewer 3 for this comment. We corrected the magnification of Figure 1.

  1. Lines 127-134 should be the part of Figure 3 description.

Reply: We thank Reviewer 3 for this advice. We adjusted the style of each Figure legend.

  1. In Figure 4 there is not necessary to provide the C and D graphs.

Reply: We thank Reviewer 3 for this comment. We deleted the Figure4C, D.

Round 2

Reviewer 2 Report

The authors responded to the comments.